# Contrastive Time Series Forecasting with Anomalies

## Abstract

Time-series forecasting predicts future values from past data. In real-world settings, some anomalous events have lasting effects and influence the forecast, while others are short-lived and should be ignored. Standard forecasting models fail to make this distinction, often either overreacting to noise or missing persistent shifts. We propose **Co-TSFA** (Contrastive Time-Series Forecasting with Anomalies), a regularization framework that learns *when to ignore anomalies and when to respond.* Co-TSFA generates input-only and input–output augmentations to model forecast-irrelevant and forecast-relevant anomalies, and introduces a latent–output alignment loss that ties representation changes to forecast changes. This encourages invariance to irrelevant perturbations while preserving sensitivity to meaningful distributional shifts. Experiments on Traffic and Electricity benchmarks, as well as on a real-world cash-demand dataset, demonstrate that Co-TSFA improves performance under anomalous conditions while maintaining accuracy on normal data. An anonymized GitHub repository with the implementation of Co-TSFA is provided here and will be made public upon acceptance.

## 1 Introduction

Time-series forecasting underpins many critical applications, including weather prediction (Nie et al., 2023), financial market modeling (Gao et al., 2024), and cash-demand forecasting for ATM replenishment (Venkatesh et al., 2014). While time series often follow regular patterns such as seasonality and trend, these patterns are frequently disrupted by anomalous events. Some anomalies cause short-term fluctuations, such as a brief surge in energy usage during a cold night, whereas others lead to persistent changes, such as the long-lasting demand shifts during the COVID-19 pandemic. These disruptions challenge forecasting models that are trained only on normal conditions.

A particularly important setting is when anomalies occur at test time, where three scenarios may arise: (i) input-only anomalies, where corrupted history should be ignored so predictions remain unaffected (Figure 1, anomalous sequence 1); (ii) anomalies that start in the input window and persist into the prediction window, where forecasts should adapt to reflect the anomaly's downstream effect (Figure 1, anomalous sequence 2); and (iii) normal conditions, where no anomaly is present and forecasts should follow the nominal trajectory. Because many anomalies are short-lived and systems can quickly return to normal, forecasting models must consistently handle all three scenarios within a single framework.

Recent forecasting models based on, for example, transformers, time-series foundation models, and self-supervised learning, such as TimesNet (Wu et al., 2023), TimeXer (Wang et al., 2025), and Autoformer (Wu et al., 2021), achieve strong performance under normal conditions but do not explicitly address test-time anomalous situations.

Non-stationary forecasting methods (Liu et al., 2023; Arik et al., 2022; Yoon et al., 2022) adapt to gradual distribution shifts but do not distinguish between transient anomalies and those with lasting effects. Existing robust forecasting approaches typically overlook anomalies that extend into the prediction horizon. Furthermore, recent robust forecasting frameworks Cheng et al. (2024) primarily address anomalies present in the training data, leaving the challenge of handling unseen anomalies at test time largely open. As a result, the problem of jointly handling normal conditions, input-only anomalies, and input–output anomalies at test time remains largely open, despite being crucial for deploying forecasting models in real-world systems.

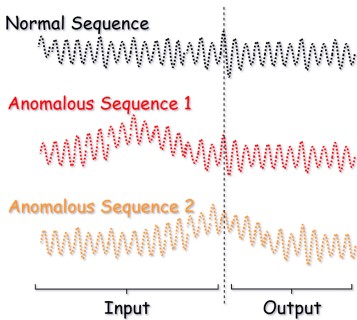

Figure 1: Illustration of anomaly types in time-series forecasting. Sequence 1 shows an input-only anomaly that should not affect the forecast, whereas Sequence 2 shows an input anomaly that persists into the output (forecast-relevant).

To address this gap, we propose **Co-TSFA**, **Co**ntrastive **T**ime-Series **F**orecasting with **A**nomalies, a framework designed to (i) suppress irrelevant input disturbances, (ii) adapt forecasts when anomalies meaningfully affect future outcomes, and (iii) preserve accuracy under normal conditions. Co-TSFA achieves this by injecting targeted synthetic anomalies during training and introducing a contrastive regularization term that aligns latent representations with forecast-relevant deviations.

Specifically, Co-TSFA computes two types of similarities: (i) between the ground-truth outputs of original and augmented samples, and (ii) between the latent representations of those same pairs. A latent–output alignment loss then minimizes the discrepancy between these similarities, ensuring that representation shifts occur only when the forecast meaningfully changes. This enforces invariance to forecast-irrelevant perturbations while maintaining sensitivity to forecast-relevant ones.

Unlike conventional robust forecasting approaches that indiscriminately suppress input variations, Co-TSFA explicitly distinguishes between anomalies that should propagate into the forecast and those that should be ignored. This enables adaptive responses to persistent regime shifts while preserving stability under transient noise.

To summarize, our main contributions are as follows:

- We formalize the problem of *Forecasting under Anomalous Conditions* by distinguishing between forecast-relevant and forecast-irrelevant anomalies and highlighting the need for representation-level guidance in this setting.

- We propose **Co-TSFA**, a contrastive regularization framework built on a taxonomy of forecast-relevant versus forecast-irrelevant augmentations, that enforces latent–output alignment, encouraging the model to respond proportionally to forecast-relevant shifts while remaining invariant to irrelevant perturbations.

- We conduct extensive experiments across multiple benchmark datasets, demonstrating that Co-TSFA improves forecasting accuracy under anomalous conditions without sacrificing performance on normal data.

## 2 Related Work

**Time-Series Forecasting under Clean Conditions.** Classical forecasting models such as ARIMA (Box & Jenkins, 2015) rely on fixed parametric and linearity assumptions, which limits their ability to capture nonlinear dynamics. Deep learning models based on RNNs, LSTMs, and GRUs relaxed these assumptions by modeling nonlinear dependencies through recurrence. The introduction of Transformers (Vaswani et al., 2017) further advanced the field by enabling efficient modeling of long-range dependencies. More recent state-of-the-art models such as Informer (Zhou et al., 2021), FEDformer (Zhou et al., 2022), iTransformer (Liu

et al., 2024), Autoformer (Wu et al., 2021), and TimeXer (Wang et al., 2025) leverage attention mechanisms to capture complex temporal dependencies, while TimesNet (Wu et al., 2023) uses frequency decomposition and convolution for efficient temporal representation learning.

**Representation Learning and Augmentations.** Learning robust representations has become a central theme in modern time-series modeling. Self-supervised and contrastive approaches such as TS2Vec (Yue et al., 2022), SoftCLT (Lee et al., 2024), and PatchTST (Nie et al., 2023) aim to learn general-purpose embeddings that transfer across tasks, while foundation-model approaches (Liang et al., 2024; Das et al., 2024; Li et al., 2025) extend this idea to large-scale cross-domain pretraining. Augmentation strategies play a key role in these methods. Policy-based approaches such as AutoAugment (Cubuk et al., 2019), RandAugment (Cubuk et al., 2020), and AutoTS-A (Yuan et al., 2025) automatically search for transformations and magnitudes, but they are not tailored to reflect meaningful anomaly patterns. Other works couple augmentations with contrastive objectives to enhance forecasting: Park et al. (Park et al., 2024) introduce an autocorrelation-based contrastive loss for long-term forecasting; CoST (Woo et al., 2022) disentangles seasonal and trend components via time- and frequency-domain contrastive objectives; and InfoTS (Luo et al., 2023) leverages information-theoretic contrastive learning to produce representations sensitive to forecast-relevant variations.

**Similarity Alignment Objectives.** The principle of aligning two similarity structures, one computed in a representation or feature space, the other derived from labels or targets, has a long history outside time-series forecasting. Kernel-target alignment Cristianini et al. (2001) measures the cosine similarity, under the Frobenius inner product, between a kernel's Gram matrix and the label-agreement matrix, and uses this quantity as a cheap surrogate for kernel and hyperparameter selection in classification, avoiding repeated cross-validation. This idea was later generalized to centered kernel alignment for kernel learning Cortes et al. (2012) and to centered kernel alignment (CKA) as a tool for comparing learned representations across networks and layers Kornblith et al. (2019). More broadly, the principle underlies much of metric learning, where a learned distance or similarity is fit to match label-derived supervision (e.g., neighborhood component analysis Goldberger et al. (2004) and large-margin nearest neighbor Weinberger & Saul (2009) ), as well as supervised contrastive learning Khosla et al. (2020), which pulls together representations of same-label pairs and pushes apart different-label pairs. This alignment principle remains active in recent representation-learning research. Target-kernel alignment continues to be studied theoretically. In large-scale self-supervised vision models, DINOv3 Siméoni et al. (2025) introduces Gram anchoring, which penalizes the discrepancy between the Gram matrix of current patch features and that of an earlier model checkpoint, to prevent dense feature degradation over long training schedules, a direct, modern instantiation of kernel/Gram-matrix alignment used as a training-time regularizer rather than as a model-selection criterion. Related analysis Insulla et al. (2025); Yeo et al. (2025) examines this family of Gram-matrix losses explicitly through the lens of kernel alignment, and concurrent work generalizes alignment-based analysis to deep representation learning more broadly. Co-TSFA belongs to this family, penalizing the discrepancy between similarities computed in latent space and over forecast targets. It differs from this prior work in what is being aligned and why. Prior alignment methods compare static similarity structures based on fixed class labels or model snapshots and serve as model selection or feature preservation criteria. Co-TSFA instead aligns changes in latent representations with changes in continuous forecast outputs induced by synthetic augmentations, making the alignment criterion dynamic, augmentation-conditioned, and specific to the forecasting setting.

**Robust and Anomaly-Aware Forecasting.** Robust forecasting methods (Yoon et al., 2022; Wang et al., 2024) mitigate input perturbations to prevent noise from propagating into predictions but generally do not reason about whether deviations are forecast-relevant. RobustTSF (Cheng et al., 2024) provides a theoretically grounded framework for robustness under contaminated training data, but it assumes clean test data and focuses on pointwise anomalies. In contrast, we study the complementary setting of clean training data with anomalous test conditions, focusing on continuous anomalies that induce prolonged distributional shifts. Together, these lines of research highlight the importance of robust representations, but they do not explicitly address the joint challenge of handling normal conditions, input-only anomalies, and input-

output anomalies at test time. Our work directly tackles this gap by training models to ignore irrelevant disturbances while adapting to anomalies that truly affect future outcomes.

## 3 Method

### 3.1 Problem Definition

We consider multivariate time-series forecasting. Each input sequence is $\mathbf{x} \in \mathbb{R}^{T \times C}$, where $T$ is the input window length and $C$ is the number of channels. The goal is to predict a future sequence $\mathbf{y} \in \mathbb{R}^{H \times C}$ over horizon $H$. The training set is $\mathcal{D}_{\text{train}} = \{(\mathbf{x}_i, \mathbf{y}_i)\}_{i=1}^{N}$ and test set is $\mathcal{D}_{\text{test}} = \{(\mathbf{x}_i, \mathbf{y}_i)\}_{i=1}^{M}$.

A forecasting model comprises an encoder $g_\phi : \mathbb{R}^{T \times C} \to \mathbb{R}^{T' \times D}$ that extracts temporal representations and a forecasting head $h_\psi : \mathbb{R}^{T' \times D} \to \mathbb{R}^{H \times C}$ that maps representations to future values. The full predictor is

$$f_\theta(\mathbf{x}) = h_\psi\big(g_\phi(\mathbf{x})\big), \qquad \theta = \{\phi, \psi\},$$

where $T'$ is the latent sequence length and $D$ is the representation dimension. The base forcasting objective minimizes a generic discrepancy $\ell(\cdot, \cdot)$ between the prediction $\hat{\mathbf{y}} = f_\theta(\mathbf{x})$ and the target $\mathbf{y}$:

$$\mathcal{L}_{\text{forecast}} = \mathbb{E}_{(\mathbf{x},\mathbf{y}) \sim \mathcal{D}_{\text{train}}}\big[\ell\big(\hat{\mathbf{y}}, \mathbf{y}\big)\big], \tag{1}$$

where $\ell$ is task-specific (e.g., MAE, MSE).

The test set may contain both normal and anomalous samples, where anomalies manifest as irregular patterns or distributional shifts in the input. Such anomalies can be short-lived, confined to the input window, or persistent, in which case their effects propagate into the prediction horizon (Fig. 1). Our goal is to develop a model that maintain performance on normal sequences while remaining robust and adaptive under anomalous test-time conditions.

### 3.2 Co-TSFA: Contrastive Time-Series Forecasting with Anomalies via Latent–Output Alignment

To improve forecasting robustness under test-time anomalies, we propose **Co-TSFA**, a contrastive regularization framework that explicitly aligns latent representations with outputs. The key assumption is that forecast-relevant shifts in the input should induce corresponding changes in the latent space, while forecast-invariant perturbations should leave the latent representation unaffected.

Co-TSFA is model-agnostic and can be applied to any forecasting model with an encoder, without modifying its architecture or primary objective. It encourages the encoder to capture forecast-relevant variations under distributional shifts, thereby improving generalization to anomalous conditions.

**Latent–Output Alignment.** Given an input–target pair $(\mathbf{x}, \mathbf{y})$, we draw an augmented pair $(\mathbf{x}', \mathbf{y}')$ from the augmentation distribution $\mathcal{A}(\mathbf{x}, \mathbf{y})$. The model encodes the inputs into latent representations $z = g_\phi(\mathbf{x})$ and $z' = g_\phi(\mathbf{x}')$, which are then decoded to produce forecasts $\hat{\mathbf{y}} = f_\theta(\mathbf{x})$ and $\hat{\mathbf{y}}' = f_\theta(\mathbf{x}')$.

The augmentation may be input-only (where $\mathbf{y}' = \mathbf{y}$) or joint input–output (where $\mathbf{y}' \neq \mathbf{y}$), thereby covering both forecast-invariant and forecast-relevant anomaly scenarios. The guiding principle is that latent representation shifts should be proportional to output shifts: if the perturbation leads to a significant output change, the latent representation should reflect this change; if the outputs remain unaffected, the latent space should remain stable.

To formalize this principle, Co-TSFA introduces a latent–output alignment loss that penalizes discrepancies between the similarity of latent representations and the similarity of their associated output. Let $\text{sim}(\cdot, \cdot)$ denote a similarity function (e.g., softmax-normalized dot product). The alignment loss is defined as

$$\mathcal{L}_{\text{align}} = \mathbb{E}_{(\mathbf{x},\mathbf{y}) \sim \mathcal{D}_{\text{train}}, \, (\mathbf{x}',\mathbf{y}') \sim \mathcal{A}(\mathbf{x},\mathbf{y})}\big[\big|\text{sim}(z, z') - \text{sim}(\mathbf{y}, \mathbf{y}')\big|\big], \tag{2}$$

where $(\mathbf{x}', \mathbf{y}')$ are augmented variants of $(\mathbf{x}, \mathbf{y})$. This loss enforces that latent representation shifts mirror output shifts, promoting invariance to irrelevant perturbations while maintaining sensitivity to meaningful

distributional changes. ==This loss is an instance of the kernel/representation-alignment family, here adapted to penalize discrepancies between augmentation-induced changes in latent space and in continuous forecast outputs, rather than between static similarity structures and fixed class labels.==

Implementing the alignment constraint requires a similarity measure that considers both positive and negative pairs. While cosine similarity or $\ell_2$ distance are possible choices, we adopt a batch-wise, softmax-normalized dot product inspired by InfoCL, which compares each representation against all other samples in the mini-batch and their augmentations. This formulation normalizes similarities relative to a set of negatives, enabling the model to learn calibrated representation shifts.

Formally, the similarity between latent representations $z$ and $z'$ at time step $t$ for the $i$-th original and augmented samples is defined as

$$\text{sim}(z_{i,t}, z'_{i,t}) = -\log \frac{\exp(z_{i,t} \cdot z'_{i,t})}{\sum_{j=1}^{B} \Big[ \exp(z_{i,t} \cdot z'_{j,t}) + \mathbb{1}[i \neq j] \exp(z_{i,t} \cdot z_{j,t}) \Big] + \sum_{k=1}^{A} \exp(z_{i,t} \cdot z'_{k,t})}, \tag{3}$$

where $B$ is the mini-batch size, $A$ is the number of augmented samples for each sequence, $z_{i,t}$ and $z'_{i,t}$ denote the latent representations at time step $t$ for the $i$-th original and augmented samples, and $\mathbb{1}[\cdot]$ is the indicator function that excludes identical pairs from the negative set. The sequence-level similarity used in Eq. 2 is obtained by aggregating the time-step similarities, i.e., $\text{sim}(z, z') = \frac{1}{T} \sum_{t=1}^{T} \text{sim}(z_{i,t}, z'_{i,t})$.

Similarly, the similarity between the ground-truth targets $\mathbf{y}$ and their augmented counterparts $\mathbf{y}'$ at time step $t$ for the $i$-th original and augmented samples is defined as

$$\text{sim}(y_{i,t}, y'_{i,t}) = -\log \frac{\exp(y_{i,t} \cdot y'_{i,t})}{\sum_{j=1}^{B} \Big[ \exp(y_{i,t} \cdot y'_{j,t}) + \mathbb{1}[i \neq j] \exp(y_{i,t} \cdot y_{j,t}) \Big] + \sum_{k=1}^{A} \exp(y_{i,t} \cdot y'_{k,t})}, \tag{4}$$

where $B$ is the mini-batch size, $A$ is the number of augmentations per sequence, $y_{i,t}$ and $y'_{i,t}$ denote the ground-truth targets at time step $t$ for the $i$-th original and augmented samples, and $\mathbb{1}[\cdot]$ excludes identical pairs from the denominator. This formulation uses the original–augmented target pairs as positives and treats all other targets and augmentations in the mini-batch as negatives. The term $\text{sim}(\mathbf{y}, \mathbf{y}')$ in Eq. 2 is computed by aggregating the per–time-step output similarities in Eq. 4. Combined with the latent-space similarity (Eq. 3), it drives the alignment loss (Eq. 2) to ensure that representations change only when the true target changes under augmentation.

**Training Objective.** Co-TSFA optimizes the forecasting model by minimizing a composite objective that couples the standard forecasting loss with the latent–output alignment regularizer. The total training loss is defined as

$$\mathcal{L}_{\text{total}} = \mathcal{L}_{\text{forecast}} + \lambda_{\text{align}} \, \mathcal{L}_{\text{align}}, \tag{5}$$

where $\mathcal{L}_{\text{forecast}}$ is the base forecasting loss (e.g., mean squared error), $\mathcal{L}_{\text{align}}$ is the alignment loss from Eq. 2, and $\lambda_{\text{align}}$ controls the trade-off between predictive accuracy and representation consistency.

This joint objective enforces two complementary properties: (i) predictive fidelity under nominal conditions through $\mathcal{L}_{\text{forecast}}$, and (ii) representation stability and proportionality under augmented scenarios through $\mathcal{L}_{\text{align}}$. As a result, the encoder learns to ignore forecast-irrelevant perturbations while remaining sensitive to input shifts that correspond to meaningful target changes.

**Overall Workflow.** The full training procedure is summarized in Fig. 2. Each mini-batch is first sampled from $\mathcal{D}_{\text{train}}$, followed by the generation of augmented pairs $(\mathbf{x}', \mathbf{y}') \sim \mathcal{A}(\mathbf{x}, \mathbf{y})$. Both original and augmented inputs are passed through the encoder $g_\phi(\cdot)$ to obtain latent representations, which are then decoded into predictions. Similarities in the latent space and target space are computed using Eqs. 3 and 4, respectively, and used to evaluate the alignment loss in Eq. 2. The model parameters $\theta = \{\phi, \psi\}$ are updated via backpropagation on the total loss (Eq. 5).

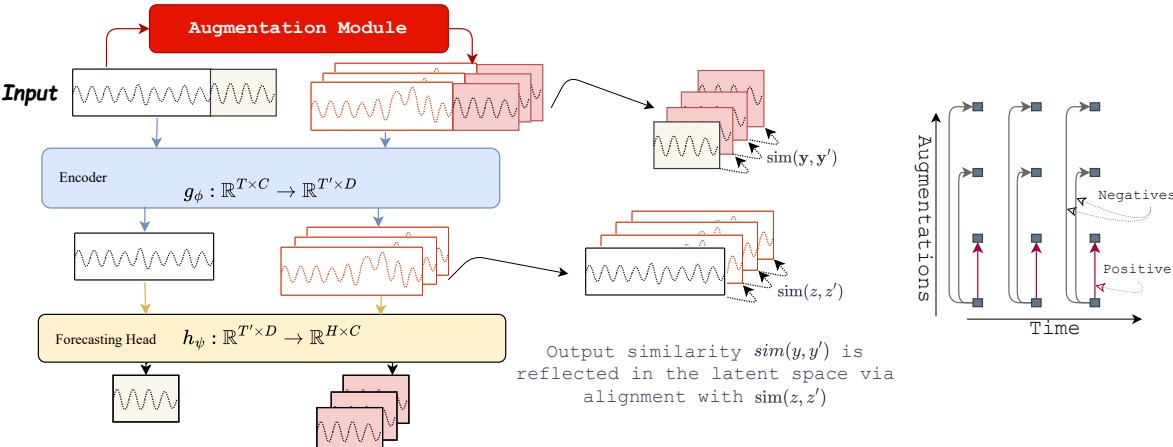

Figure 2: (Left) pipeline of Co-TSFA. (Right)positive and negative pairs.

This design ensures that Co-TSFA does not simply maximize latent invariance, but learns a calibrated response where representation shifts are aligned with forecast-relevant changes in the ground truth.

### 3.3 Augmentations

A central component of Co-TSFA is the construction of augmented pairs that induce controlled distributional shifts, providing the supervision signal required for latent–output alignment. By adding such perturbations during training, Co-TSFA gives the model practice through supervised contrastive learning with realistic anomaly patterns, helping it handle similar situations more effectively when they occur at test time. Unlike conventional contrastive frameworks that perturb only the input $\mathbf{x}$, Co-TSFA applies augmentations either to $\mathbf{x}$ alone or jointly to $(\mathbf{x}, \mathbf{y})$.

Input-only augmentation captures variations that do not affect the forecasting target, as illustrated by anomalous situation 2 in Fig. 1. The model is encouraged to remain invariant to such perturbations, promoting robustness to forecast-irrelevant noise. For a time series sequence

$$\mathbf{x} = \{x_1, x_2, \ldots, x_L\}, \quad \mathbf{y} = \{y_1, \ldots, y_P\},$$

with encoder length $L$ and prediction length $P$, we define an anomaly curve:

$$a(t) = \frac{A \cdot t \cdot \exp\left(-B \cdot t^C\right)}{Z}, \quad t \geq 0,$$

where $A, B, C$ are sampled anomaly parameters shared across the batch (with small per-sample Gaussian variation), and $Z$ is a scaling constant. This anomaly function was extracted with non-linear regression for an anomalous event. Therefore, it is directly based on anomalies that may appear in real-world data.

The perturbed sequence is then

$$\tilde{x}_t = x_t + \mu(\mathbf{x}, \mathbf{y}) \cdot a(t - t_0), \quad t \geq t_0,$$

where $\mu(\mathbf{x}, \mathbf{y})$ denotes the mean value of the sequence and $t_0 \in [0, 0.5L]$ is the anomaly start index for the input-only anomalies.

This type of anomaly affects only the encoder input, while the forecast horizon remains clean. It simulates situations where historical data is corrupted (e.g., logging errors or short-lived disturbances), but the actual future is unaffected.

Conversely, input–output augmentation represents persistent or structural anomalies where shifts in $\mathbf{x}$ must propagate to $\mathbf{y}$, necessitating a forecast adjustment. This contrasts with conventional robust forecasting,

which would treat such shifts as noise and suppress them. In contrast, Input+Output anomalies are injected so that they overlap both the encoder and the prediction horizon. The anomaly start point is chosen late in the input window: $t_0 \in [0.85L, 0.95L]$, ensuring that the perturbation extends into the forecast window.

The perturbed sequence is given by

$$\tilde{x}_t = \begin{cases} x_t + \mu(\mathbf{x}, \mathbf{y}) \cdot a(t - t_0), & t \leq L, \\ y_{t-L} + \mu(\mathbf{x}, \mathbf{y}) \cdot a(t - t_0), & L < t \leq L + P, \end{cases}$$

so that both the tail of the input and the forecast horizon is affected by the anomaly.

This case is more challenging for the forecasting model, as it must capture *anomalous future dynamics* rather than reverting to a normal trend. It reflects real-world crisis scenarios (e.g., panic-driven cash withdrawals) where the abnormal behavior is not confined to the past but extends into the future.

By combining these two augmentation modes, Co-TSFA explicitly encodes the inductive bias that latent representations should be invariant to irrelevant perturbations but sensitive to forecast-relevant ones, ensuring that representation shifts align only with meaningful changes in the predicted output.

## 4 Experiments

**Datasets.** We evaluate our method on the benchmarks: the *Traffic* dataset[1] and the *Electricity* dataset[2], both widely used in time-series forecasting, as well as a real-world *Cash Demand* dataset. The details are in Appendix A.1. We further include an additional experiment on the ETTh1 benchmark (Zhou et al., 2021) using the iTransformer backbone, with full results and statistical analysis reported in Appendix A.5.

**Evaluation metrics.** For the cash demand dataset, we evaluate using the mean absolute error (MAE), mean squared error (MSE), and symmetric mean absolute percentage error (SMAPE). Since this dataset contains many zeros due to ATM downtime and service intervals, and SMAPE is highly sensitive to zeros, we compute SMAPE only on timesteps with strictly positive true values. This adjustment prevents unpredictable downtime periods from dominating the error measure, while MAE and MSE are reported on the full series. For the data sets of traffic and electricity, we follow the evaluation protocol of (Cheng et al., 2024) and report only MAE and MSE.

**Training Details.** All models are implemented in PyTorch and run on single- or dual-GPU setups with Nvidia RTX 4000 Ada Generation 20GB cards. We train all models using the Adam optimizer with an initial learning rate of 0.001 or 0.0001. A step-decay learning rate schedule halves the learning rate after every epoch. The batch size is set to 128. Training runs for 2 epochs on the cash demand dataset, 4 epochs on the traffic and electricity datasets with our setup, and 10 epochs on the traffic and electricity datasets with the setup from RobustTSF(Cheng et al., 2024), with early stopping patience of 3 based on validation loss. We evaluate our method using fixed input/output lengths: 128/64 for all experiments on the Cash Demand dataset, and 16/1 for all experiments on the Traffic and Electricity datasets, following the RobustTSF protocol. All input series are normalized using statistics from the training split. For Co-TSFA, we combine the standard forecasting loss (MSE) with the latent–output alignment loss, weighted by $\lambda_{CL} = 0.1$. Each input sequence is augmented 5 times using both input-only and input–output anomaly injection strategies. Unless otherwise stated, results are averaged over 3 random seeds (0, 1, 2).

**Anomaly types.** Our primary focus is on *continuous anomalies*, representing prolonged deviations from normal behavior. Specifically, we study *input-only anomalies* (Anomalous Sequence 1 in Fig. 1) and *input-output anomalies* (Anomalous Sequence 2 in Fig. 1). We evaluate robustness under such continuous anomalous conditions in two settings: long-term forecasting on the cash demand dataset (Table 1) and single-step forecasting on the traffic dataset (Table 2). As mentioned in subsection 3.2, we sample anomalies from an anomaly function. For this anomaly function, we fix $B = 0.385$ and $Z = 90,409$, while sampling the

---

[1]https://pems.dot.ca.gov/
[2]https://archive.ics.uci.edu/dataset/321/electricityloaddiagrams20112014

other parameters from Gaussian distributions: $A \sim \mathcal{N}(74{,}120, 20{,}000^2)$, and $C \sim \mathcal{N}(0.806, 0.2^2)$. To ensure realistic dynamics, we impose additional constraints: the anomaly curve must remain non-negative, have a maximum value below 2.0, and stay below 0.4 at day 30. These conditions prevent extreme outliers and ensure that the injected perturbations stay within reasonable limits.

In addition, we compare against the robust forecasting approach in RobustTSF (Cheng et al., 2024) that targets *pointwise anomalies*. Pointwise anomalies appear as isolated spikes or drops at random time steps. Following the characterization in RobustTSF, we adopt three types: constant, missing, and Gaussian. We use their default parameterization, with anomaly scale 0.5 for constant anomalies and 2.0 for Gaussian anomalies, and we evaluate across anomaly ratios $\{0.1, 0.2, 0.3\}$.

### 4.1 Results

#### 4.1.1 Main Results

We begin with the primary setting considered throughout this paper, in which the training data contains only normal sequences, whereas the test data may include anomalous sequences. We evaluate three scenarios: (i) **Clean**, where no anomalies are present, which serves as a reference to verify that adding Co-TSFA does not degrade normal forecasting performance; (ii) **Input-Only**, where anomalies are confined to the input window and should ideally be ignored to prevent forecast deviations; and (iii) **Input+Output**, where anomalies span both the input and output windows, requiring the model to adapt its predictions to reflect the underlying regime shift.

We compare six state-of-the-art forecasting models (TimesNet (Wu et al., 2023), TimeXer (Wang et al., 2025), Autoformer (Wu et al., 2021), Informer (Zhou et al., 2021)), PAttn (Tan et al., 2024), iTransformer (Liu et al., 2024) with and without the proposed Co-TSFA regularization. Results on the *Cash Demand* dataset are summarized in Table 1. Each cell reports the mean performance across three seeds. For each condition (Clean, Input-Only, Input+Output), we show the baseline performance, the performance after adding Co-TSFA, and the relative improvement $\Delta = \frac{\text{Error}_{\text{Co-TSFA}} - \text{Error}_{\text{base}}}{\text{Error}_{\text{base}}} \times 100\%$, where negative values indicate improvement (lower error) and zero means no change. To aid interpretation, $\Delta \leq 0$ values are highlighted in red.

Overall, we observe three consistent trends: (i) performance degrades across all models when anomalies are introduced, with the largest degradation occurring in the input–output setting where anomalies propagate into the prediction window; (ii) incorporating Co-TSFA consistently reduces MAE, MSE, and SMAPE across all models, with the largest gains observed in the input–output setting, confirming its ability to both adapt forecasts to regime shifts and improve relative calibration under distributional shifts.

All baseline models benefit from the addition of Co-TSFA. For the remaining experiments, we adopt TimesNet as the representative backbone for all subsequent experiments and use it to systematically evaluate the effect of Co-TSFA under a broader range of anomaly settings.

#### 4.1.2 Different anomaly types in test data

To assess whether the robustness learned by Co-TSFA generalizes beyond its training augmentation family, we evaluate the model under a diverse set of unseen test-time anomaly types. Following the taxonomy of RobustTSF Cheng et al. (2024), we consider continuous anomalies (input-only and input+output) together with pointwise constant, missing, and Gaussian corruptions at multiple severity levels. Since Co-TSFA is trained exclusively with continuous parametric anomalies, the pointwise settings constitute an out-of-distribution evaluation of its robustness.

Table 2 reports the results on the *Traffic* dataset. Co-TSFA consistently achieves lower MAE and MSE than RobustTSF on clean data, continuous anomalies, and the constant and missing pointwise anomaly settings across all evaluated corruption ratios. The largest improvement is observed for continuous input+output anomalies, where Co-TSFA reduces the MAE from 0.8647 to 0.2064, demonstrating its ability to maintain accurate forecasts under sustained distribution shifts. Moreover, performance degrades gracefully as

Table 1: Forecasting performance on the **Cash Demand** dataset. MAE ↓, MSE ↓, SMAPE ↓ (mean over 3 seeds). Δ shows relative improvement (%) of +Co-TSFA over baseline (negative = better). Red indicates equal or improved performance.

| Model | Metric | Clean | | | Input-Only | | | Input+Output | | |
|---|---|---|---|---|---|---|---|---|---|---|
| | | Base | +Co-TSFA | Δ% | Base | +Co-TSFA | Δ% | Base | +Co-TSFA | Δ% |
| TimesNet | MAE | 0.232 | **0.231** | -0.4 | 0.276 | **0.268** | -2.9 | 0.369 | **0.357** | -3.2 |
| | MSE | **0.159** | **0.159** | 0.0 | 0.202 | **0.195** | -3.5 | 0.367 | **0.350** | -4.6 |
| | SMAPE | **28.98** | 29.07 | +0.3 | 29.38 | **29.02** | -1.2 | 34.17 | **31.24** | -8.6 |
| PAttn | MAE | **0.255** | 0.258 | +1.2 | **0.269** | 0.274 | +1.9 | 0.341 | **0.324** | -5.0 |
| | MSE | **0.185** | 0.187 | +1.1 | **0.202** | 0.208 | +3.0 | 0.316 | **0.279** | -11.7 |
| | SMAPE | **31.72** | 31.80 | +0.3 | **33.87** | 34.21 | +1.0 | 40.84 | **40.17** | -1.6 |
| TimeXer | MAE | **0.265** | **0.265** | 0.0 | 0.276 | **0.275** | -0.4 | 0.324 | **0.321** | -0.9 |
| | MSE | **0.199** | **0.199** | 0.0 | 0.212 | **0.210** | -0.9 | 0.279 | **0.277** | -0.7 |
| | SMAPE | 31.78 | **31.65** | -0.4 | 33.74 | **33.66** | -0.2 | 40.24 | **39.85** | -1.0 |
| iTransformer | MAE | **0.237** | 0.238 | +0.4 | 0.253 | **0.251** | -0.8 | 0.359 | **0.3328** | -7.3 |
| | MSE | **0.165** | **0.165** | 0.0 | 0.183 | **0.180** | -1.6 | 0.350 | **0.276** | -21.1 |
| | SMAPE | **30.00** | 30.08 | +0.3 | 32.31 | **32.15** | -0.5 | 41.59 | **39.47** | -5.1 |
| Autoformer | MAE | 0.304 | **0.290** | -4.6 | 0.328 | **0.324** | -1.2 | 0.332 | **0.317** | -4.5 |
| | MSE | 0.221 | **0.214** | -3.2 | **0.250** | 0.251 | +0.4 | 0.264 | **0.254** | -3.8 |
| | SMAPE | 35.72 | **34.07** | -4.6 | 38.46 | **36.72** | -4.5 | 42.58 | **40.93** | -3.9 |
| Informer | MAE | 0.275 | **0.270** | -1.8 | **0.276** | **0.276** | 0.0 | **0.301** | **0.301** | 0.0 |
| | MSE | 0.208 | **0.203** | -2.4 | **0.210** | 0.217 | +3.3 | **0.248** | **0.248** | 0.0 |
| | SMAPE | 36.60 | **33.67** | -8.0 | 37.70 | **35.26** | -6.5 | 41.04 | **40.41** | -1.5 |

the severity of constant and missing pointwise anomalies increases, indicating that robustness learned from continuous anomaly augmentation transfers effectively to several unseen pointwise corruption types.

The only exception is Gaussian pointwise anomalies, where RobustTSF consistently achieves lower errors. This behavior is expected given the different design objectives of the two methods. RobustTSF is specifically optimized for pointwise corruptions during training, whereas Co-TSFA is trained exclusively with temporally continuous parametric anomalies and is never exposed to isolated Gaussian spikes. Consequently, Gaussian pointwise anomalies represent a natural cross-family generalization limitation for Co-TSFA. Nevertheless, its strong performance across continuous anomalies and the remaining pointwise settings demonstrates that the learned representations generalize beyond the augmentation family used during training rather than overfitting to specific anomaly patterns.

### 4.1.3 Anomalous training data

All results so far assumed clean training data, isolating robustness to test-time anomalies. In practice, however, training data may be collected during disrupted regimes such as pandemics, supply chain failures, or sensor malfunctions. To investigate this substantially more challenging setting in which both the training and test data are contaminated, with different anomaly families. Unlike the previous experiment, this experiment assesses not only robustness to test-time perturbations but also the ability of the learned representations to remain effective when the training distribution is itself corrupted. We consider both continuous and pointwise anomalies during training and evaluate the models under matched and mismatched train-test contamination patterns across the Traffic and Electricity datasets. We inject anomalies into the training data under two regimes: *continuous* contamination, where entire temporal segments are shifted, and *pointwise* contamination, where individual samples are corrupted. These regimes mimic persistent regime shifts and localized faults, respectively, allowing us to examine whether models can still learn meaningful representations under corrupted supervision.

We evaluate both RobustTSF and Co-TSFA on the *Traffic* and *Electricity* datasets, using clean, input-only, input+output, and pointwise-corrupted test data with varying anomaly types (constant, missing, Gaussian)

Table 2: Forecasting performance on the **Traffic** dataset with clean training data. This setup isolates robustness to test-time anomalies. Severity indicates the proportion of points affected in pointwise anomalies. Best values per row are **bold**.

| Anomaly Class | Anomaly Type | Ratio (%) | RobustTSF | | Co-TSFA (Ours) | |
|---|---|---|---|---|---|---|
| | | | MAE ↓ | MSE ↓ | MAE ↓ | MSE ↓ |
| none | clean | – | 0.1927 | 0.1099 | **0.1545** | **0.0572** |
| continuous | input-only | – | 0.5135 | 0.5481 | **0.1862** | **0.0731** |
| continuous | input+output | – | 0.8647 | 1.3267 | **0.2064** | **0.0876** |
| pointwise | const | 10 | 0.2580 | 0.1592 | **0.2202** | **0.0988** |
| pointwise | const | 20 | 0.3083 | 0.1943 | **0.2700** | **0.1337** |
| pointwise | const | 30 | 0.3448 | 0.2321 | **0.3039** | **0.1577** |
| pointwise | missing | 10 | 0.3884 | 0.3919 | **0.3165** | **0.2222** |
| pointwise | missing | 20 | 0.5333 | 0.6121 | **0.4182** | **0.3385** |
| pointwise | missing | 30 | 0.6045 | 0.6641 | **0.4814** | **0.4212** |
| pointwise | gaussian | 10 | **0.4244** | **0.6174** | 0.5119 | 0.7466 |
| pointwise | gaussian | 20 | **0.6318** | **1.1399** | 0.8216 | 1.5396 |
| pointwise | gaussian | 30 | **0.7872** | **1.5621** | 1.0647 | 2.2862 |

and severity levels. When two perturbations are listed in the "Test Perturbation(s)" column, the first corresponds to a continuous anomaly and the second to a pointwise anomaly applied jointly. Table 3 report MAE and MSE across all combinations of training and test contamination.

When the test data contain continuous anomalies, Co-TSFA consistently achieves substantially lower errors than RobustTSF on both datasets, regardless of whether the training data are clean, continuously contaminated, or corrupted by pointwise anomalies. The performance gap is particularly pronounced for input+output anomalies, where RobustTSF exhibits severe error escalation while Co-TSFA maintains comparatively low forecasting errors. These results indicate that the representations learned through continuous anomaly augmentation transfer effectively even when the training distribution is partially corrupted.

For pointwise test anomalies, the behavior is more nuanced. Co-TSFA consistently outperforms RobustTSF on constant and, in most cases, missing corruptions, especially when the training contamination differs from the test anomaly type, suggesting improved robustness under distribution mismatch. In contrast, Gaussian pointwise anomalies remain the primary failure mode of Co-TSFA. RobustTSF generally achieves lower errors in these settings, particularly on the Traffic dataset, which is consistent with its design objective of explicitly modeling pointwise corruptions during training. Nevertheless, Co-TSFA remains competitive on several Gaussian settings, particularly on the Electricity dataset and under continuous training contamination, indicating that its learned robustness extends beyond the anomaly family used during training.

Taken together, these results demonstrate that Co-TSFA is resilient to training-time contamination, even though it was not explicitly designed to learn from corrupted supervision. Remarkably, it outperforms RobustTSF in many scenarios, despite RobustTSF being specifically developed for robustness to contaminated training data. This suggests that the representations learned by Co-TSFA capture a broader notion of robustness that generalizes across diverse combinations of training- and test-time anomalies. The primary exception remains Gaussian pointwise anomalies, where RobustTSF's targeted design provides a clear advantage.

For completeness, Appendix A.2 evaluates the case where the training data are contaminated with anomalies while the test data remain clean.

Table 3: Forecasting performance on the **Traffic** and **Electricity** datasets with contaminated train and test data. The first four columns describe the training/test setup. The "Train Contam." column refers to the type of anomalies injected into the training data, while the "Test Contam." column refers to the type of anomalies injected into the test data. Results are grouped by dataset, with MAE↓/MSE↓ reported for both RobustTSF (R) and Co-TSFA (O). Best results per row are **bold**.

| Train Contam. | Test Contam. | Anomaly Type | Ratio (%) | Traffic | | | | Electricity | | | |
|---|---|---|---|---|---|---|---|---|---|---|---|
| | | | | MAE (R) | MSE (R) | MAE (O) | MSE (O) | MAE (R) | MSE (R) | MAE (O) | MSE (O) |
| Continuous | none | input-only, – | – | 0.2029 | 0.1101 | **0.1633** | **0.0592** | **0.1825** | **0.0700** | 0.1997 | 0.0826 |
| | none | input+output, – | – | 0.2152 | 0.1184 | **0.1824** | **0.0722** | 0.2184 | 0.0995 | **0.2149** | **0.0928** |
| | continuous | input-only, – | – | 0.3956 | 0.4022 | **0.1862** | **0.0731** | 0.4517 | 0.8015 | **0.2340** | **0.1121** |
| | continuous | input+output, – | – | 0.8665 | 1.4120 | **0.2064** | **0.0876** | 2.7849 | 10.6588 | **0.3940** | **0.2887** |
| | pointwise | input-only, const | 10 | 0.2748 | 0.1679 | **0.2247** | **0.0997** | 0.2588 | 0.1317 | **0.2584** | **0.1281** |
| | pointwise | input-only, const | 30 | 0.3467 | 0.2234 | **0.3065** | **0.1580** | 0.3408 | 0.1991 | **0.3318** | **0.1842** |
| | pointwise | input-only, missing | 10 | 0.4050 | 0.4397 | **0.3124** | **0.2123** | **0.3243** | 0.2838 | 0.3280 | 0.2269 |
| | pointwise | input-only, missing | 30 | 0.6394 | 0.8116 | **0.4724** | **0.4067** | 0.4637 | 0.4834 | **0.4473** | **0.3851** |
| | pointwise | input-only, Gaussian | 10 | **0.4472** | **0.6727** | 0.4977 | 0.7026 | **0.4283** | **0.6375** | 0.5088 | 0.6925 |
| | pointwise | input-only, Gaussian | 30 | **0.8245** | **1.5619** | 1.0120 | 2.1323 | **0.8136** | **1.6215** | 0.9535 | 1.9319 |
| | pointwise | input+output, const | 10 | 0.3007 | 0.1985 | **0.2385** | **0.1097** | 0.3031 | 0.1805 | **0.2680** | **0.1343** |
| | pointwise | input+output, const | 30 | 0.3894 | 0.2811 | **0.3107** | **0.1629** | 0.3721 | 0.2409 | **0.3329** | **0.1859** |
| | pointwise | input+output, missing | 10 | 0.4395 | 0.4985 | **0.3196** | **0.2153** | 0.3744 | 0.2943 | **0.3218** | **0.2325** |
| | pointwise | input+output, missing | 30 | 0.6327 | 0.6664 | **0.4826** | **0.4155** | 0.4516 | 0.3977 | **0.4553** | **0.3942** |
| Pointwise | none | const, – | 10 | 0.1982 | 0.1133 | **0.1675** | **0.0629** | 0.1754 | **0.0663** | 0.2062 | 0.0877 |
| | none | const, – | 30 | 0.2029 | 0.1126 | **0.1805** | **0.0700** | **0.1864** | **0.0710** | 0.2175 | 0.0947 |
| | none | missing, – | 10 | **0.1930** | 0.1078 | 0.2093 | **0.0882** | **0.1810** | **0.0704** | 0.2442 | 0.1151 |
| | none | missing, – | 30 | **0.2346** | 0.1401 | 0.2982 | 0.1490 | **0.1907** | **0.0745** | 0.3205 | 0.1802 |
| | none | Gaussian, – | 10 | **0.1904** | **0.1096** | 0.2468 | 0.1117 | **0.1788** | **0.0689** | 0.2540 | 0.1223 |
| | none | Gaussian, – | 30 | **0.1959** | **0.1095** | 0.3544 | 0.1956 | **0.1771** | **0.0663** | 0.4060 | 0.2632 |
| | continuous | input-only, const | 10 | 0.4734 | 0.4603 | **0.1875** | **0.0748** | 0.5149 | 0.9640 | **0.2284** | **0.1075** |
| | continuous | input-only, const | 30 | 0.5039 | 0.5631 | **0.1968** | **0.0795** | 0.4991 | 0.9831 | **0.2339** | **0.1109** |
| | continuous | input-only, missing | 10 | 0.4441 | 0.4403 | **0.2241** | **0.0966** | 0.5241 | 0.9479 | **0.2488** | **0.1221** |
| | continuous | input-only, missing | 30 | 0.5202 | 0.5671 | **0.2903** | **0.1430** | 0.5277 | 1.0025 | **0.3014** | **0.1640** |
| | continuous | input-only, Gaussian | 10 | 0.4321 | 0.4127 | **0.2548** | **0.1179** | 0.5251 | 1.0936 | **0.2856** | **0.1502** |
| | continuous | input-only, Gaussian | 30 | 0.4572 | 0.4833 | **0.3483** | **0.1942** | 0.5052 | 0.9295 | **0.3549** | **0.2142** |
| | continuous | input+output, const | 10 | 0.9538 | 1.5124 | **0.1996** | **0.0825** | 2.3452 | 7.7038 | **0.3823** | **0.2682** |
| | continuous | input+output, const | 30 | 0.9767 | 1.4631 | **0.2204** | **0.0946** | 2.2127 | 6.9110 | **0.3965** | **0.2839** |
| | continuous | input+output, missing | 10 | 0.9848 | 1.5439 | **0.2701** | **0.1325** | 2.0848 | 6.4235 | **0.4199** | **0.3121** |
| | continuous | input+output, missing | 30 | 0.9985 | 1.5450 | **0.3433** | **0.2028** | 2.1099 | 6.5936 | **0.4871** | **0.4067** |
| | continuous | input+output, Gaussian | 10 | 0.8537 | 1.3724 | **0.3046** | **0.1611** | 2.0781 | 6.5427 | **0.4747** | **0.3781** |
| | continuous | input+output, Gaussian | 30 | 0.9126 | 1.4337 | **0.4011** | **0.2649** | 2.0146 | 6.3299 | **0.7898** | **0.9431** |

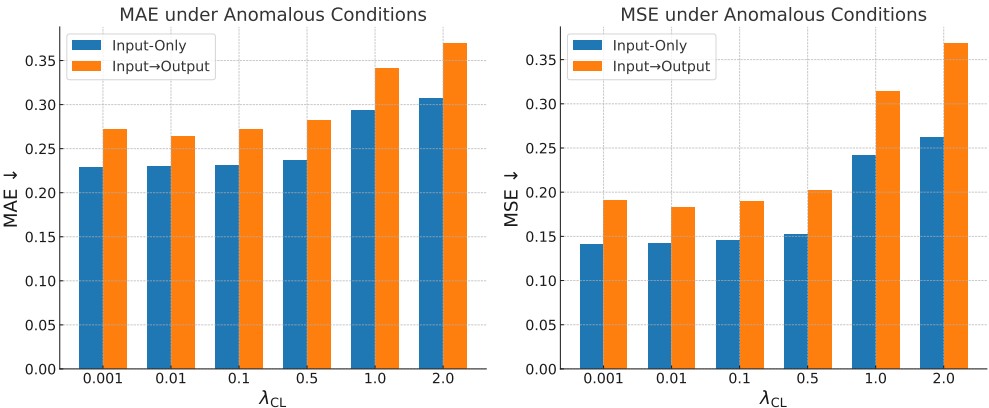

Figure 3: Effect of $\lambda_{\mathrm{CL}}$ on MAE↓ (left) and MSE↓ (right) under anomalous conditions for input-only and input-to-output scenarios.

**Effect of Co-TSFA Weight.** We analyze the effect of the Co-TSFA regularization weight $\lambda_{\mathrm{CL}}$ on forecasting performance. We sweep $\lambda_{\mathrm{CL}} \in \{0.001, 0.01, 0.1, 0.5, 1.0, 2.0\}$ and evaluate under anomalous conditions for both (i) input-only and (ii) input–output anomaly scenarios. Figure 3 reports MAE and MSE as a function of $\lambda_{\mathrm{CL}}$. Key observations are as follows: (i) performance remains stable for small $\lambda_{\mathrm{CL}} \leq 0.1$, (ii) moderate regularization ($\lambda_{\mathrm{CL}} \approx 0.1$–$0.5$) provides the best trade-off, improving anomalous-case accuracy, and

(iii) large weights ($\lambda_{\mathrm{CL}} \geq 1.0$) degrade performance, suggesting that excessive contrastive pressure harms representation quality.

## 5 Conclusion

This paper introduced **Co-TSFA**, a contrastive regularization framework for improving the robustness of time-series forecasting models under anomalous conditions. By generating input-only and input–output augmentations and enforcing a latent–output alignment loss, Co-TSFA learns to ignore forecast-irrelevant perturbations while adapting to forecast-relevant anomalies. Experiments on benchmark and real-world datasets demonstrate that Co-TSFA improves forecasting accuracy under anomalous conditions without sacrificing clean-data performance.

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

# A Appendix

## A.1 Experimental setup

**Datasets.** The cash demand dataset consists of daily aggregated withdrawals from 1,323 ATMs from 1 January 2023 to 31 October 2024. Each timestep has three columns: the date, ATM-id (unique ID for each ATM), and the total amount that was withdrawn from the specific ATM-id during that date. In total, the data set contains 616501 time steps, meaning that some ATMs do not have data for the entire period.

## A.2 Generalization to clean test after anomalous training.

We consider the regime in which the training data are contaminated with anomalies while the test set remains clean. This setting assesses whether models trained under disrupted conditions can recover normal-regime performance at test time. Results are reported in Table 4.

Table 4: Traffic dataset: Training data contains pointwise anomalies, test data are clean. This is the main setup of RobustTSF. We report MAE↓ and MSE↓ for RobustTSF and Co-TSFA (Ours). Best values per row are **bold**.

| Anomaly Type | Ratio | RobustTSF | | Co-TSFA (Ours) | |
|---|---|---|---|---|---|
| | | MAE | MSE | MAE | MSE |
| clean | – | 0.1927 | 0.1099 | **0.1545** | **0.0572** |
| const | 0.1 | 0.1982 | 0.1133 | **0.1675** | **0.0629** |
| | 0.2 | 0.2049 | 0.1147 | **0.1745** | **0.0667** |
| | 0.3 | 0.2029 | 0.1126 | **0.1805** | **0.0700** |
| missing | 0.1 | **0.1930** | 0.1078 | 0.2093 | **0.0882** |
| | 0.2 | 0.2236 | 0.1357 | **0.2504** | **0.1147** |
| | 0.3 | 0.2346 | 0.1401 | **0.2982** | **0.1490** |
| gaussian | 0.1 | **0.1904** | **0.1096** | 0.2468 | 0.1117 |
| | 0.2 | **0.1886** | **0.1004** | 0.3057 | 0.1571 |
| | 0.3 | **0.1959** | **0.1095** | 0.3544 | 0.1956 |

## A.3 Comparison with Fine-Tuning

In this section, we compare Co-TSFA with standard fine-tuning. Fine-tuning continues training on data containing anomalies, but does not distinguish forecast-relevant from irrelevant deviations. As shown in Table 5, this leads to severe performance degradation under clean conditions, suggesting overfitting to anomalous patterns. Co-TSFA, in contrast, preserves accuracy on clean data while improving robustness to input-only anomalies and adapting forecasts when anomalies affect future values. This highlights that naive fine-tuning on anomalous data is insufficient for reliable forecasting.

Table 5: Forecasting performance of TimesNet variants on the **Cash Demand** dataset. MAE ↓, MSE ↓, SMAPE ↓ (mean over 3 seeds). The Fine-tuned result on clean data is based on the model that was fine-tuned on input-output anomalies.

| Model | Metric | Clean | | | Input-Only | | | Input+Output | | |
|---|---|---|---|---|---|---|---|---|---|---|
| | | Base | Fine-tuned | Co-TSFA | Base | Fine-tuned | Co-TSFA | Base | Fine-tuned | Co-TSFA |
| TimesNet | MAE | 0.232 | 0.504 | **0.231** | 0.276 | 0.279 | **0.268** | 0.369 | **0.289** | 0.357 |
| | MSE | 0.159 | 0.614 | **0.159** | 0.202 | 0.207 | **0.195** | 0.367 | **0.220** | 0.350 |
| | SMAPE | **28.98** | 48.05 | 29.07 | 29.38 | 30.27 | **29.02** | 34.17 | 32.93 | **31.24** |

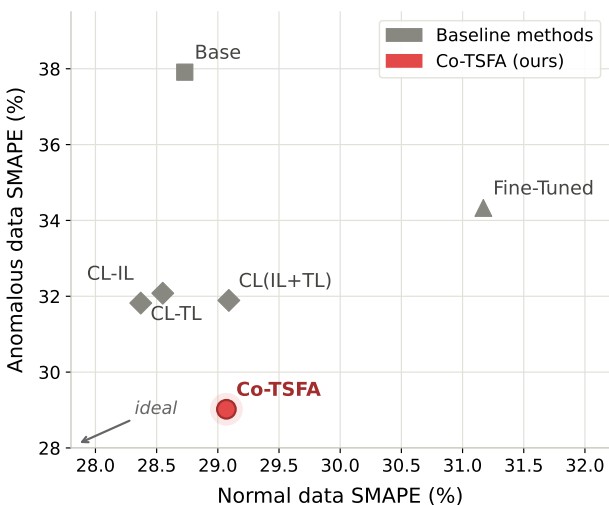

Figure 4: Trade-off between normal and anomalous data SMAPE (%) for all ablation variants. Co-TSFA (red) achieves the best anomalous-data performance while preserving clean-data accuracy, in contrast to fine-tuning which improves anomalous performance at the cost of normal data degradation. Bottom-left is ideal.

## A.4 Comparison with Contrastive Learning Baselines

To further situate Co-TSFA within the contrastive learning literature for time series, we compare against TS2Vec Yue et al. (2022), one of the closest contrastive learning approaches for time-series representation learning. TS2Vec learns universal representations via two complementary objectives: instance-wise contrastive learning, which encourages representations of the same sequence under different augmentations to be similar, and temporal contrastive learning, which enforces consistency across time steps within a sequence. We compare against all variants of this framework: instance-level only (CL-IL), temporal-level only (CL-IL), and both combined CL(IL+TL), alongside a normally trained baseline, a fine-tuned variant, and Co-TSFA with input-only augmentations. Results are reported in Table 6 in terms of SMAPE (%) on normal data (ND) and anomalous data (AD) using TimesNet as the backbone. Co-TSFA achieves the best performance under anomalous conditions (29.02 vs. 37.91 for the normally-trained baseline and 31.82 for the best TS2Vec variant), while preserving clean data accuracy (29.07 vs. 28.73 for the base model), a negligible difference that confirms Co-TSFA does not hurt normal forecasting performance. To further illustrate this, Figure 4 visualizes the trade-off between normal and anomalous data performance for all variants.

Table 6: Ablation study on TimesNet (SMAPE %). ND: Normal Data, AD: Anomalous Data.

| Data | TimesNet (Base) | Fine-Tuned | Contrastive Learning | | | CO-TSFA |
| --- | --- | --- | --- | --- | --- | --- |
| | | | CL(IL+TL) | CL-TL | CL-IL | |
| Clean Data | 28.73 | 31.17 | 29.09 | **28.37** | 28.55 | 29.07 |
| Anomalous Data | 37.91 | 34.33 | 31.89 | 31.82 | 32.08 | **29.02** |

## A.5 Additional Results on ETTh1 dataset

We report additional results on the ETTh1 dataset using iTransformer as the backbone. As in the main experiments, we consider three regimes: *Clean*, *Input-Only* (anomalies injected only in the input history), and *Input+Output* (anomalies injected in both the input history and the prediction window). For each regime, we compare the base iTransformer with the same model trained using the proposed Co-TSFA regularization (denoted "+Co").

Table 7 summarizes the performance in terms of MAE, MSE, and SMAPE, reported as mean (standard deviation) over 15 seeds. On clean data, Co-TSFA achieves performance comparable to the base model, with a slight increase in MAE, MSE and SMAPE. Under the more challenging *Input-Only* and *Input+Output* anomaly settings, Co-TSFA consistently improves all metrics over the base iTransformer, with particularly pronounced gains in the heavily perturbed *Input+Output* case. **These results support our claim that Co-TSFA enhances robustness to anomalous conditions without sacrificing performance on clean data.**

Table 7: Forecasting performance on the ETTh1 dataset using iTransformer. Mean (standard deviation) over 15 runs; lower is better. "+Co" denotes the addition of Co-TSFA (history=128, horizon=64).

| Metric | Clean | Clean+Co | Input-Only | Input-Only+Co | In+Out | In+Out+Co |
|---|---|---|---|---|---|---|
| MAE | **0.225** (0.002) | 0.229 (0.004) | 0.242 (0.004) | **0.236** (0.003) | 0.323 (0.012) | **0.293** (0.009) |
| MSE | **0.089** (0.001) | 0.090 (0.002) | 0.104 (0.004) | **0.098** (0.003) | 0.179 (0.014) | **0.149** (0.002) |
| SMAPE | **17.442** (0.831) | 17.936 (0.897) | 16.801 (0.883) | **16.148** (0.672) | 16.390 (0.608) | **15.583** (0.457) |

To assess the statistical reliability of these differences, we perform paired t-tests comparing each regime with and without Co-TSFA. That is, we evaluate *Clean vs Clean+Co*, *Input-Only vs Input-Only+Co*, and *Input+Output vs Input+Output+Co*. Table 8 summarizes the results using the conventions: ✓✓ ($p < 0.01$), ✓ ($p < 0.05$), and - (not significant). The analysis shows that the improvements introduced by Co-TSFA are statistically significant or highly significant across all metrics and settings, without sacrificing performance on clean data.

Table 8: Paired t-test significance comparing the base iTransformer with its Co-TSFA variant under each anomaly regime. ✓✓: $p < 0.01$, ✓: $p < 0.05$, –: not significant.

| Metric | Clean vs Clean+Co | Input-Only vs Input-Only+Co | In+Out vs In+Out+Co |
|---|---|---|---|
| MAE | – | ✓ | ✓✓ |
| MSE | – | ✓ | ✓✓ |
| SMAPE | – | ✓ | ✓✓ |

We further analyze the robustness of Co-TSFA on the *ETTh1* dataset by changing the history and horizon length, and considering history = 96, horizon = 24 by repeating each configuration over 15 independent runs. Table 9 reports the mean and standard deviation for MAE, MSE, and SMAPE. The resulting t-statistics and $p$-values are summarized in Table 10. Results use the standard significance conventions: $p < 0.01$ (✓✓), $0.01 \leq p < 0.05$ (✓), and non-significant otherwise.

Table 9: Forecasting performance on the ETTh1 dataset using iTransformer. Mean (standard deviation) over 15 runs; lower is better. "+Co" denotes the addition of Co-TSFA (history=96, horizon=24).

| Metric | Clean | Clean+Co | Input-Only | Input-Only+Co | In+Out | In+Out+Co |
|---|---|---|---|---|---|---|
| MAE | **0.164** (0.050) | 0.166 (0.050) | 0.179 (0.060) | **0.174** (0.056) | 0.288 (0.157) | **0.278** (0.141) |
| MSE | **0.089** (0.010) | 0.091 (0.002) | 0.060 (0.004) | **0.058** (0.002) | 0.157 (0.015) | **0.140** (0.014) |
| SMAPE | 17.266 (0.382) | **17.033** (0.575) | 11.662 (0.348) | **10.605** (0.375) | 16.240 (0.985) | **15.605** (0.973) |

Table 10: Paired t-test significance comparing the base iTransformer with its Co-TSFA variant under each anomaly regime. ✓✓: $p < 0.01$, ✓: $p < 0.05$, –: not significant.

| Metric | Clean vs Clean+Co | Input-Only vs Input-Only+Co | In+Out vs In+Out+Co |
|---|---|---|---|
| MAE | – | ✓✓ | – |
| MSE | – | ✓✓ | ✓✓ |
| SMAPE | ✓✓ | ✓✓ | ✓ |

## A.6 Significance Analysis Under the Input+Output Anomaly Regime

To isolate the effect of Co-TSFA specifically under the most challenging anomaly regime, *Input+Output* corruption, we extract all performance values corresponding to this setting and compare different backbone model against their Co-TSFA-enhanced variants. The Input+Output case represents the scenario in which both the historical inputs and the forecasting targets are perturbed, making it the regime where robustness is most critical. For each backbone and metric (MAE, MSE, SMAPE), we report the mean and standard deviation computed over 15 independent runs. The final column indicates the paired t-test significance level of the improvement (or degradation) when moving from the base model to its Co-TSFA variant. Table 11 summarizes the results.

Table 11: Performance in the Input+Output anomaly regime (mean ± std over 15 runs). Lower is better. "Sig." denotes paired t-test significance comparing In+Out vs In+Out+Co.

| Model | Metric | In+Out | In+Out+Co | Sig. |
|---|---|---|---|---|
| iTransformer | MAE | 0.347 (0.024) | **0.319** (0.020) | ✓✓ |
| iTransformer | MSE | 0.330 (0.040) | **0.278** (0.031) | ✓✓ |
| iTransformer | SMAPE | 41.261 (1.156) | **39.568** (1.224) | ✓✓ |
| PAttn | MAE | 0.338 (0.048) | **0.324** (0.049) | ✓ |
| PAttn | MSE | 0.309 (0.047) | **0.281** (0.039) | ✓✓ |
| PAttn | SMAPE | 40.803 (1.395) | **40.375** (1.431) | – |
| TimeXer | MAE | 0.328 (0.024) | **0.306** (0.022) | ✓✓ |
| TimeXer | MSE | 0.288 (0.031) | **0.253** (0.032) | ✓✓ |
| TimeXer | SMAPE | 39.893 (1.213) | **38.786** (1.102) | ✓✓ |

## A.7 Training Dynamics and Loss Curves

To provide additional transparency regarding optimization behavior and model stability, we report the training dynamics of Co-TSFA on the CashDemand dataset. Figure 5 shows the evolution of the Co-TSFA alignment loss, while Figure 6 presents the forecasting losses (MSE) for both the training and validation sets.

Across training, the Co-TSFA loss decreases smoothly and stabilizes without oscillation, indicating that the alignment objective is well-behaved and does not introduce training instability. The forecasting losses exhibit a similarly stable downward trend, and the validation loss follows the training loss closely without divergence, suggesting that Co-TSFA does not cause overfitting or optimization drift.

These results further confirm that the proposed method integrates cleanly into standard forecasting pipelines and maintains stable learning dynamics.

## A.8 Additional Evidence of Irregularity in the ATM Transaction Data

To further support our claim that the CashDemand (ATM) dataset exhibits highly irregular and non-stationary behavior, we provide additional visualizations of transaction volumes from two randomly selected ATMs. Unlike standard forecasting benchmarks that display clear seasonal cycles or predictable temporal structures, the ATM series shown in Figures 7 reveal strong volatility, abrupt spikes, and inconsistent fluctuation patterns.

Both ATMs exhibit rapid changes in transaction levels, with large amplitude variations and sharp jumps that do not repeat periodically. These characteristics highlight the absence of stable seasonality and the presence of machine-specific dynamics, making this dataset distinctly challenging. This further confirms that evaluating robustness under anomaly-induced perturbations are particularly relevant in this real-world setting.

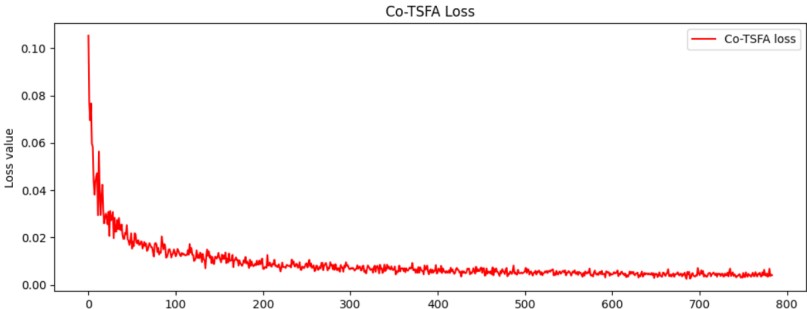

Figure 5: Evolution of the Co-TSFA alignment loss during training. Each step on the x-axis corresponds to 10 batches.

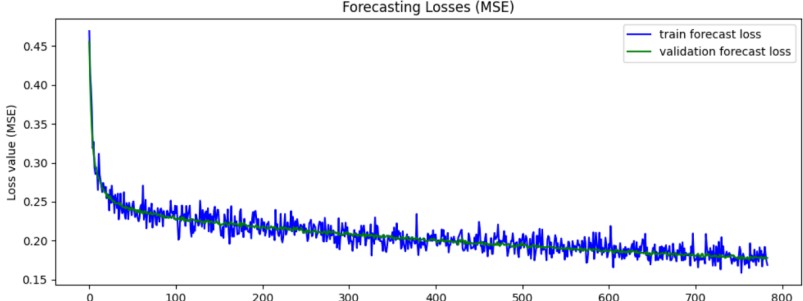

Figure 6: Training and validation forecasting losses (MSE) over training. The curves demonstrate stable optimization without divergence.

### A.9 Use of Large Language Models

A large language model was employed only for spelling, grammar, and clarity of phrasing. It played **no** role in developing ideas, designing methods, implementing experiments, analyzing data, or reporting results. All scientific content, interpretations, and conclusions are entirely written by the paper's authors.

### A.10 Discussion

**Assumptions underlying the anomaly generation process.** The anomaly curve used in Co-TSFA was obtained via non-linear regression fitted to real anomalous events observed in the cash demand dataset, capturing the characteristic shape of real-world disruptions: a rapid rise followed by a gradual decay back to baseline. The parameters A, B, C are sampled from Gaussian distributions with constraints (non-negativity, maximum value below 2.0, value below 0.4 at day 30) to prevent extreme outliers while preserving realistic diversity across training samples.

The key assumptions are: (i) anomalies have a unimodal, transient shape, they rise and decay rather than inducing permanent step changes; (ii) their amplitude scales with the local mean of the series, making the injection magnitude adaptive to each sequence; and (iii) The boundary between input-only and input+output anomalies is determined by the anomaly start time relative to the forecast horizon.

These assumptions are reasonable in domains where disruptions are event-driven and self-correcting. In ATM cash demand, panic-driven withdrawal surges revert to baseline within days; in traffic, incident-driven congestion dissipates once resolved; in energy consumption, weather-driven spikes decay as temperatures normalize. All three domains share a characteristic rise-and-decay structure that the parametric curve captures well. The assumptions are less likely to hold when anomalies manifest as permanent structural breaks, regulatory changes that shift demand indefinitely, factory closures that remove a persistent load, or rerouted traffic patterns that never revert. Such settings would be better served by step-function augmentations that do not decay. Oscillatory or recurring anomalies similarly fall outside the unimodal assumption and would require a

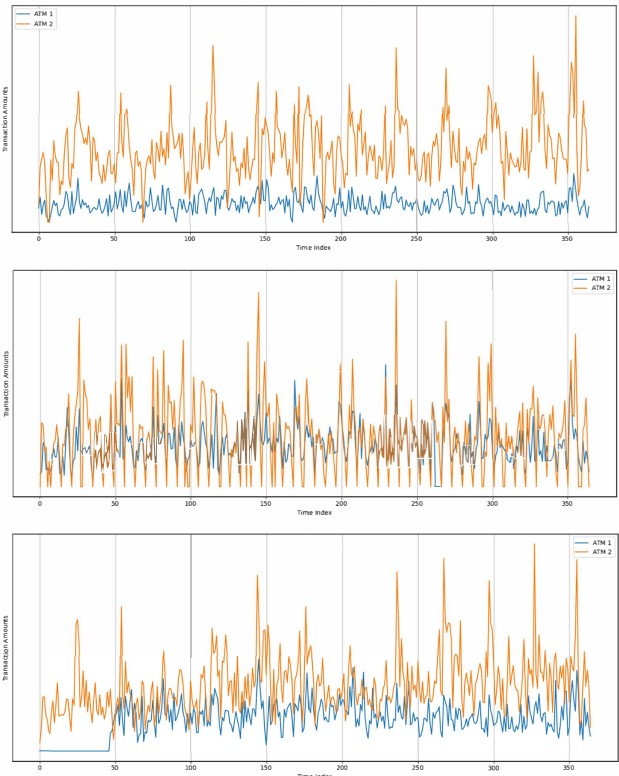

Figure 7: ATM transaction volumes over different time slots for randomly selected ATMs, showing high volatility, sharp spikes, and a lack of repeating temporal structure.

different augmentation family. This points to an important design principle: the anomaly generation process is an inductive bias that should be informed by domain knowledge. Practitioners are encouraged to inspect historical anomalous events and fit the anomaly curve to representative examples, as done here. Where such characterization is unavailable, the cross-family robustness demonstrated in Tables 2 and 3 suggests that Co-TSFA still generalizes meaningfully beyond its training anomaly family.

**Is Co-TSFA forecasting-specific or a general framework?** The underlying principle aligning representation shifts with target shifts under augmentation is general, and we have now acknowledged this explicitly in the revised Related Work and after Eq. 2. However, the instantiation of this principle in Co-TSFA is fundamentally forecasting-specific in two ways that cannot be trivially transferred to other settings:

First, the augmentation taxonomy itself depends on the existence of a forecast horizon. The distinction between input-only and input+output anomalies is defined by whether the injected perturbation crosses the boundary between the input window and the prediction window. This boundary does not exist in classification or general representation learning, making the taxonomy a concept native to forecasting.

Second, the alignment target is a continuous, time-varying forecast output rather than a discrete class label or a fixed model snapshot. This requires the similarity measure (Eqs. 3–4) to operate over sequences rather than scalar labels, and makes the alignment criterion sensitive to the magnitude and temporal structure of forecast deviations.

## A.11  Broader Impact

Co-TSFA is relevant for forecasting in high-stakes domains such as energy, traffic, finance, and cash-demand management, where the cost of misclassifying an anomaly as irrelevant, or vice versa, can be significant. A model that incorrectly ignores a persistent regime shift may underreact to a real crisis, while one that overreacts to transient noise may trigger unnecessary interventions. We therefore caution that Co-TSFA

should not be deployed as a standalone decision system in safety-critical environments. Responsible deployment should include anomaly monitoring, uncertainty estimation, human oversight, and rollback procedures. Furthermore, the synthetic augmentation family used during training may not cover all anomaly patterns encountered in practice. We recommend incorporating domain expert knowledge to define anomaly profiles, complemented by domain-specific validation on real anomalous data, before operational use.

