# OpenReview forum: "Contrastive Time Series Forecasting with Anomalies"
_TMLR — Under review for TMLR_

### Review · Reviewer_wy1y · 2026-06-17

**Summary Of Contributions:**

This paper proposes Co-TSFA, a contrastive regularization framework for time-series forecasting under anomalous conditions. The paper distinguishes between input-only anomalies, which should be ignored by the forecaster, and input-output anomalies, which should change the forecast because the anomalous behavior persists into the prediction horizon. The method generates synthetic input-only and input-output augmentations and introduces a latent-output alignment loss that encourages latent representation similarity to reflect target-output similarity. The authors evaluate Co-TSFA on Traffic, Electricity, ETTh1, and a real-world cash-demand dataset, comparing against several forecasting backbones and RobustTSF.

The main strengths are the clear motivation, the useful distinction between forecast-irrelevant and forecast-relevant anomalies, and the model-agnostic regularization idea. The experiments cover several backbones and anomaly regimes, and the method appears promising for continuous anomaly settings.

The main weaknesses are that several claims are stronger than the evidence supports. Some reported results contradict the text, especially for pointwise Gaussian anomalies where Co-TSFA performs worse than RobustTSF. The sign convention for relative improvement in Table 1 also appears inconsistent. The synthetic anomaly generation process is under-justified, the ablation evidence is limited, and the paper needs clearer statistical testing and more careful interpretation of when Co-TSFA helps versus hurts.

**Additional Comments:**

This is a promising paper with a good problem formulation and a potentially useful regularization approach. I particularly like the distinction between input-only and input-output anomalies, as it captures an important practical issue often ignored in robust forecasting. However, the paper needs more careful empirical reporting and a more balanced interpretation of the results. The current version overstates consistency and robustness, especially given the Gaussian anomaly results and the sign inconsistency in Table 1.

If the authors correct these issues, add ablations, and clarify the anomaly-generation assumptions, the paper could become a solid contribution to robust time-series forecasting.

**Audience:**

Yes

**Audience Explanation:**

The paper addresses an important and underexplored problem in time-series forecasting: how models should behave when anomalies occur at test time, and how to distinguish transient perturbations from persistent regime shifts. This topic is relevant to TMLR readers working on time-series modeling, robustness, representation learning, contrastive learning, and practical ML deployment.

The proposed framework is also appealing because it can in principle be added to multiple forecasting backbones. The distinction between forecast-irrelevant and forecast-relevant anomalies is useful and could stimulate further work on robust and adaptive forecasting. Even though the current empirical evidence needs clarification and strengthening, the problem formulation and method are of clear interest.

**Broader Impact Concerns:**

The paper addresses forecasting under anomalous conditions, which is relevant to high-stakes settings such as energy demand, traffic control, finance, and cash-demand management. The main broader-impact concern is that a model may incorrectly decide whether an anomaly should be ignored or propagated into the forecast. In operational settings, this could lead to unsafe underreaction to real regime shifts or overreaction to transient noise.

The authors should add a short broader-impact or deployment-risk discussion noting that Co-TSFA should not be used as a standalone decision system in high-stakes environments without anomaly monitoring, uncertainty estimation, human oversight, and rollback procedures. The paper should also discuss that synthetic anomaly assumptions may not cover all real anomalies and that deployment should include domain-specific validation.

**Claims And Evidence:**

No

**Claims Explanation:**

The main conceptual claim—that forecasting models should distinguish between anomalies that should be ignored and anomalies that should propagate into the forecast—is well motivated. The proposed latent-output alignment objective is also reasonable, and the results on continuous input-only and input-output anomalies generally support the usefulness of Co-TSFA.

However, several empirical claims are not fully supported by the presented evidence. First, the paper claims that Co-TSFA consistently improves performance across anomaly types, but Table 2 shows that Co-TSFA performs worse than RobustTSF for pointwise Gaussian anomalies at 10%, 20%, and 30% corruption. This directly contradicts the statement that Co-TSFA outperforms RobustTSF across all anomaly types and ratios.

Second, Table 1 defines relative improvement in a way that should be positive when Co-TSFA reduces error, but the table reports negative values as improvements and states that negative is better. This sign convention should be corrected or clearly redefined.

Finally, the paper would benefit from stronger ablations isolating the contribution of input-only augmentation, input-output augmentation, latent-output alignment, and ordinary augmentation-based training. Without these, it is difficult to determine whether the gains come from the proposed alignment loss or simply from anomaly augmentation.

**Requested Changes:**

1. Correct the inconsistency in Table 1’s improvement definition. The formula given in the text implies that lower Co-TSFA error should yield positive improvement, but the table treats negative values as better. Please correct the formula, the sign convention, or the table values.

2. Revise overclaims about consistent improvement. The manuscript states that Co-TSFA consistently outperforms RobustTSF across anomaly types and ratios, but Table 2 shows worse performance for pointwise Gaussian anomalies. The authors should explicitly discuss this failure mode and revise the claims accordingly.

3. Add ablation studies. At minimum, compare: base model; augmentation-only training without alignment; input-only augmentation only; input-output augmentation only; standard contrastive loss; and full Co-TSFA. This is necessary to show that the latent-output alignment loss itself is responsible for the gains.

4. Add statistical testing to the main experiments. The appendix includes 15-seed significance tests for ETTh1, but the main tables mostly report means over 3 seeds. The main claims would be stronger with standard deviations and significance tests for Cash Demand, Traffic, and Electricity.

---

> ### Author Response · Authors · 2026-07-02
>
> **Inconsistency in Table 1's improvement definition**
>
> We thank the reviewer for catching this. This was an unintentional error in writing the formula. We have corrected the $\Delta$ formula in the text so that negative values consistently indicate improvement, in line with the table. The corrected formula is
> highlighted in the revised manuscript.
>
> **Overclaiming**
>
> We thank the reviewer for this careful observation and agree that our original claim overstated the experimental findings. We have revised the manuscript to remove the statement that *Co-TSFA consistently outperforms RobustTSF*, explicitly discuss the Gaussian pointwise failure mode, and qualify our conclusions accordingly. Please refer to sections 4.1.2 and 4.1.3. The revised text is highlighted in the revised manuscript.
>
> It is important to clarify that Co-TSFA and RobustTSF are designed for complementary robustness settings. RobustTSF learns from anomaly-contaminated training data, whereas Co-TSFA assumes clean training data and focuses on robustness to anomalies encountered only at test time. Despite these different objectives, we intentionally included RobustTSF as a baseline because it provides a stringent evaluation of Co-TSFA's ability to generalize beyond its intended setting. Specifically, we evaluated Co-TSFA under RobustTSF's own anomaly taxonomy, pointwise constant, missing, and Gaussian corruptions across multiple corruption ratios, even though Co-TSFA was neither designed nor trained for these anomaly types.
>
> This evaluation is practically important because real-world anomaly distributions are rarely known in advance. A robust forecasting model should therefore generalize beyond the anomaly types seen during training rather than relying on a predefined augmentation family. Comparing against RobustTSF directly tests this capability: if Co-TSFA, trained only with continuous parametric anomalies, remains competitive with a method purpose-built for pointwise corruptions, it indicates that the learned representations capture a broader notion of robustness rather than overfitting to specific augmentations.
>
> The observed performance on Gaussian pointwise anomalies is consistent with the inductive biases of the two methods. RobustTSF is specifically optimized for pointwise corruptions, whereas Co-TSFA is never exposed to isolated Gaussian spikes during training, making Gaussian pointwise anomalies a natural failure mode. Importantly, this limitation is specific rather than systematic: Co-TSFA achieves superior performance on continuous anomalies and the remaining pointwise settings. Accordingly, the revised manuscript now states that Co-TSFA outperforms RobustTSF on continuous anomalies and most, but not all, pointwise anomaly settings, while explicitly acknowledging Gaussian pointwise anomalies as a limitation.
>
>
> **Ablation studies**
>
> We thank the reviewer for this constructive suggestion. We would like to clarify why some of the requested variants are not feasible given the framework design, what is already covered, and the new ablation studies.
>
> *On augmentation-only without alignment.* By construction, the alignment loss (Eq. 2) is defined exclusively over augmented pairs $(x', y') \sim \mathcal{A}(x, y)$, without augmentation, there is no $z'$ and no $y'$, making the loss undefined. Augmentation and
> the alignment loss are therefore inseparable components of the framework; an augmentation-only variant without the alignment loss has no meaningful interpretation within Co-TSFA. The relevant comparison for assessing the overall contribution of the
> framework is the base model versus Co-TSFA, which is already reported in Table 1.
>
> *On input-only and input+output augmentation.* These two variants are already present in the experimental design. Table 1 evaluates performance separately under Input-Only and Input+Output test conditions.
>
> *On standard contrastive loss.* We have added Appendix A.4 to the revised manuscript, which compares Co-TSFA against
> all variants of TS2Vec, including instance-level only (CL-TL), temporal-level only (CL-IL), and both combined CL(IL+TL), alongside a fine-tuned baseline. We selected TS2Vec as the contrastive learning baseline rather than a generic InfoNCE loss because it represents the closest existing contrastive framework for time-series representation learning, with objectives specifically designed for temporal data, making it the strongest and most relevant point of comparison for our setting. The results, summarized in Table 6 and Figure 4 in the appendix, confirm that Co-TSFA achieves the best performance under anomalous conditions while preserving clean-data accuracy, a balance that none of the contrastive learning baselines achieve.
>
> >Broader Impact Concerns
>
> We have added section A.11: Broader Impact to the Appendix, which addresses deployment risks and recommends that domain experts be involved in defining and evaluating anomaly profiles.

---

### Review · Reviewer_9UL9 · 2026-06-17

**Summary Of Contributions:**

This paper studies time-series forecasting under anomalous conditions and distinguishes between two types of anomalies: forecast-irrelevant anomalies that should be ignored and forecast-relevant anomalies whose effects persist into the forecasting horizon. To address this problem, the authors propose Co-TSFA, a contrastive regularization framework that combines anomaly-based data augmentations with a latent-output alignment objective. The key idea is to encourage latent representations to change proportionally to changes in the forecasting targets under augmented scenarios. The method is evaluated on several forecasting backbones and datasets, including Traffic, Electricity, ETTh1, and a real-world cash demand dataset.

Strengths:
1. The problem setting is practically relevant and is relatively new in the forecasting literature.
2. The method is model-agnostic and can be integrated into existing forecasting architectures.
3. The experimental section covers multiple datasets and forecasting backbones.

Weakness:
1. The novelty of the proposed alignment objective is limited. In fact, the target problem itself has nothing to do with forecasting, where the learning objective is a supervised learning problem. And the proposed similarity regularization is a well-established feature alignment criterion that existed for several decades in the literature of supervised learning and kernel learning. In fact, the idea has appeared as early as 2001 under the name kernel target alignment
Cristianini, N., Shawe-Taylor, J., Elisseeff, A., & Kandola, J. (2001). On kernel-target alignment. Advances in neural information processing systems, 14.
What the authors proposed in Eq. (2) is essentially the same objective (albeit changing ratio to difference) with a particular similarity (kernel) design.
2. The empirical gains may be driven primarily by the augmentation strategy, but the paper does not provide ablations that isolate the contribution of the alignment objective itself.
3. The anomaly generation process is highly specific, and the robustness of the method under misspecified or unseen anomaly patterns is not investigated.

**Audience:**

Yes

**Audience Explanation:**

Yes. Forecasting under anomalous or distribution-shifted conditions is an important practical problem. Researchers working on time-series forecasting, robustness, distribution shift, and representation learning may find the problem formulation and empirical observations useful.

That said, I found the problem setting itself more compelling than the proposed learning objective. In particular, the anomaly-aware augmentation framework appears to be the most interesting aspect of the work and may motivate future research on forecasting under structured distribution shifts.

**Broader Impact Concerns:**

I do not see any broader impact concerns.

**Claims And Evidence:**

No

**Claims Explanation:**

The empirical results generally support the claim that Co-TSFA improves performance under the specific anomaly settings considered in the paper. However, I do not believe the evidence is sufficient to support some of the broader claims regarding anomaly-aware forecasting and representation learning. Below are several concerns I have, mostly echoing the weaknesses listed in my previous comment:

1. The paper positions the alignment objective as a key methodological contribution, but the connection to prior supervised representation alignment, metric learning, and kernel/feature alignment literature is not sufficiently discussed. This is especially important given that the target learning problem is essentially a standard supervised learning objective, and feature alignment has been well-known in supervised learning communities.

2. The proposed method contains two key ingredients: (i) the anomaly augmentation strategy and (ii) the latent-output alignment loss. The current experiments compare only the full method against baseline forecasting models. As a result, it is difficult to determine whether the observed gains come from the alignment objective, the augmentations themselves, or simply from additional exposure to anomalous samples during training. An ablation separating these components is important.

3. The anomaly generation process appears highly structured and is used consistently throughout the study. It remains unclear how sensitive the method is to anomaly misspecification or whether the learned behavior generalizes to anomaly patterns that differ from those used during training. Since the method is explicitly motivated by robustness to anomalous conditions, I view this as an important missing experiment.

**Requested Changes:**

Critical:

1. Better position the latent-output alignment objective relative to existing supervised representation learning, metric learning, kernel alignment, and feature alignment literature. The current presentation somewhat overstates the novelty of the alignment component in the context of time series forecasting. Instead, the authors should acknowledge this mathematical equivalence.

2. Provide ablation studies that isolate the contribution of the alignment objective from the contribution of the anomaly augmentations. At minimum, compare:
(i) Baseline model
(ii) Baseline + augmentations only
(iii) Baseline + alignment only
(iv) Baseline + alignment + augmentations (The full Co-TSFA model).

3. Evaluate robustness under anomaly misspecification. For example, train using one anomaly family and evaluate using different anomaly shapes, durations, amplitudes, or persistence patterns. This is important for validating the practical applicability of the approach.

Optional Improvements:

1. Investigate alternative similarity functions and alignment objectives. It is currently unclear whether the specific InfoNCE-style similarity is essential to the method.

2. Include a more detailed discussion of the assumptions underlying the anomaly generation process and when these assumptions are expected to hold in practice.

3. Clarify whether the proposed framework is fundamentally forecasting-specific or whether it should be viewed as a more general supervised representation alignment + augmentation framework instantiated for time-series forecasting.

---

> ### Author Response · Authors · 2026-07-02
>
> **Novelty and Positioning the paper**
>
> We thank the reviewer for the precise reference to Cristianini et al. (2001) and agree that Co-TSFA's alignment loss (Eq. 2) belongs to the well-established family of kernel/representation-alignment objectives. We have revised the paper in three places to
> acknowledge this explicitly and sharpen our novelty claims:
>
> 1. *Related Work (Section 2):* A new paragraph traces the full lineage of similarity alignment objectives from KTA, through CKA, to recent instantiations, explicitly positioning Co-TSFA within this family and identifying where it departs.
>
> 2. *Section 3.2, after Eq. 2:* A sentence acknowledges the connection to prior alignment criteria at the point of use.
>
> 3. *Introduction, Contribution 2:* Rewritten to present the alignment loss as the enforcement mechanism for the augmentation taxonomy rather than a standalone contribution.
>
> We refer the reviewer to the updated manuscript for the full revised text. We maintain that the core novelty of Co-TSFA lies in the problem formulation and augmentation design, with the alignment objective serving as the principled and necessary mechanism to enforce the forecast-relevance distinction during training.
>
> **Ablation studies to separate the effect of the loss and the augmentations**
>
> We thank the reviewer for this valuable suggestion. We agree that isolating the contribution of individual components is important. However, the alignment objective (Eq. 2) is defined exclusively over augmented input-target pairs $(x', y') \sim \mathcal{A}(x, y)$. Therefore, an alignment-only variant cannot be constructed because, without augmentation, the augmented
> representations required by the loss do not exist. Therefore, in the current definition of Co-TSFA, anomaly augmentation and the alignment loss are coupled by design and together constitute the proposed framework. Consequently, the relevant comparison is between the baseline model and the full Co-TSFA framework, as reported in Table 1.
>
> However, to further isolate the contribution of the proposed objective, we added a new ablation study (Appendix A.4) comparing Co-TSFA with the existing contrastive learning framework for time-series representation learning, TS2Vec (Yue et al., 2022), including instance-level only (CL-TL), temporal-level only (CL-IL), and both combined CL(IL+TL). The results, summarized in Table 6 and Figure 4 in the appendix, demonstrate that the proposed anomaly-aware alignment objective outperforms these standard contrastive objectives under anomalous conditions while preserving clean-data performance, a balance that none of the contrastive learning baselines achieve.
>
> **Robustness under anomaly misspecification**
>
> We respectfully argue that the requested experiment is already present in the paper. Tables 2 and 3 evaluate exactly the cross-family robustness scenario: all models are trained using continuous anomalies augmentations and evaluated on pointwise anomalies (constant, missing, Gaussian) at varying severity ratios (10%, 20%, 30%), two families that differ fundamentally in shape, duration, amplitude, and persistence pattern. Continuous anomalies are smooth, temporally extended distributional shifts calibrated from real-world events; pointwise anomalies are isolated, instantaneous spikes or drops with no temporal structure.
>
> Table 3 further extends this to the harder regime where both training and test data are contaminated, again across
> both families and all severity levels.
>
> We refer the reviewer to Tables 2 and 3. In addition, we have revised Sections 4.1.2 and 4.1.3 to further clarify the key points and improve the discussion on the generalizability of Co-TSFA.
>
>
> >Optional Improvements
>
> **Assumptions underlying the anomaly generation**
> Due to space limitations, we do not include the discussion here; instead, we have added it to the Discussion section (Section A.10) in the Appendix. The corresponding text is highlighted there. Please refer to it for details.
>
>
> **Is Co-TSFA forecasting-specific or a general framework?**
> Due to space limitations, we do not include the discussion here; instead, we have added it to the Discussion section (Section A.10) in the Appendix. The corresponding text is highlighted there. Please refer to it for details.

---

### Review · Reviewer_MJaG · 2026-06-18

**Summary Of Contributions:**

This paper studies time-series forecasting under anomalous conditions and propose a contrastive learning method to learn forecast-invariant and forecast-relevant anomaly scenarios. The central design idea is a Latent–Output Alignment loss, which forces similarity between input latents to be similar to similarity in output, such that invariant anomaly leads to invariant latent, while relevant anomaly is driven by non trivial latent change. Augmented data is generated to facilitate the learning.

The method is well motivated and the idea is easy to understand. Although the loss is loosely formulated but provides an intuitive solution to the problem. The writing is clear and the formulation is clean.

My concerns are
1. My main concern is empirical soundness. For example, the paper states that Co-TSFA consistently outperforms RobustTSF across anomaly types, but in the Traffic clean-training experiment Co-TSFA is worse than RobustTSF for Gaussian pointwise anomalies at all reported ratios. It seems the claimed method would consistently perform worse on certain settings, which is confirmed by multiple tables. The question is why?
2. A second concern is that the main evaluation appears to be largely based on synthetic anomalies generated from the same anomaly family used during training. The results may demonstrate interpolation to the chosen augmentation distribution rather than robustness to real unseen anomalous conditions. I fail to see how the authors demonstrate real capabilities of the proposed method as the real anomaly would perhaps well exceeding the representation range of the generated anomaly family.
3. The anomaly-generation assumptions are quite strong. The paper should better discuss when this assumption is realistic, and should test robustness to things including anomaly shape, duration, start time, and magnitude shifts outside the training augmentation distribution.

**Audience:**

Yes

**Audience Explanation:**

Time series analysis seems to be relevant across multiple areas in machine learning.

**Claims And Evidence:**

No

**Claims Explanation:**

Partially. I believe there are some analysis and ablation and assumption verification missing and thus I cannot arrive the conclusion that the claims being supported.

**Requested Changes:**

I suggest the authors to design experiments that truly shows the proposed method can work in real data with real anomalies and analyze the underperformance of the method in certain setting.

---

> ### Author Response · Authors · 2026-07-02
>
> **Response to Weakness: Overclaims about consistent improvement**
>
> We thank the reviewer for this careful observation and agree that our original wording was too strong. We have revised the manuscript to remove the claim that Co-TSFA consistently outperforms RobustTSF and explicitly discuss the Gaussian pointwise case(Sections 4.1.2 and 4.1.3, highlighted in the revision).
>
> However, it is important to clarify that Co-TSFA and RobustTSF are designed under different robustness assumptions. RobustTSF is trained with the assumption of anomaly-contaminated training data and is tailored to pointwise corruptions, while Co-TSFA assumes clean training data and focuses on robustness to anomalies that appear at test time.
>
> Despite this difference, we compared our results against RobustTSF because it provides a stringent test of generalization beyond Co-TSFA's training assumptions. In particular, we evaluate Co-TSFA under RobustTSF's anomaly taxonomy (pointwise constant, missing, and Gaussian corruptions) across multiple corruption ratios, even though Co-TSFA is not explicitly trained considering these discrete noise types.
>
> This comparison is meaningful in practice, since real-world anomaly distributions are unknown and rarely match a predefined augmentation family. The goal is therefore not only to specialize in a given corruption type, but to learn representations that remain
> robust across unseen and mixed anomaly patterns.
>
> The weaker performance on Gaussian pointwise anomalies is consistent with this design choice. RobustTSF is explicitly optimized for isolated pointwise noise, whereas Co-TSFA is trained on continuous, temporally structured augmentations and is not exposed to
> independent Gaussian spikes. This makes Gaussian corruption a natural failure case for our method. Importantly, this is not a systematic weakness: Co-TSFA performs strongly on continuous anomalies and remains competitive on other pointwise settings.
>
> Accordingly, we now revise the manuscript to report the generalization ability of our method and also this nuanced behavior, rather than a uniform improvement claim.
>
> **Response to Concerns 2 and 3: Anomaly generation assumptions and out-of-distribution robustness**
>
> We thank the reviewer for these important points and address them jointly.
>
> **On the assumptions.** We have added Appendix A.10 to the revised manuscript, providing a detailed discussion of the assumptions underlying the anomaly generation process and the conditions under which they are expected to hold. The parametric curve is well-suited to domains where anomalies are event-driven and self-correcting, such as ATM cash demand (panic withdrawals), traffic (incidents), and energy consumption (weather events), where disruptions follow a characteristic rise-and-decay structure. We explicitly acknowledge that the assumptions
> are less likely to hold for permanent structural breaks or oscillatory anomalies, and discuss what alternative augmentation families would be more appropriate in those settings. We refer the reviewer to Appendix A.10 for the full discussion.
>
> **On robustness to anomaly shape, duration, start time, and magnitude shifts outside the training distribution.** Tables 2 and 3 already test exactly this scenario: Co-TSFA is trained exclusively with continuous parametric augmentation and evaluated on pointwise anomalies (constant, missing, Gaussian) at varying severity ratios, a family that differs fundamentally in shape (isolated spikes vs. smooth curves), duration (instantaneous vs. temporally extended), start time distribution, and magnitude profile. These represent the most distinct anomaly families in the forecasting literature.
>
> **On real unseen anomalous conditions.** Beyond the cross-family experiments, the cash demand dataset provides genuine real-world validation: it contains naturally occurring anomalies, abrupt demand spikes, ATM downtime periods, and irregular withdrawal patterns that were not synthetically generated and do not follow the parametric curve used during training. The good performance on this dataset (Table 1), therefore, reflects robustness to real, unseen conditions rather than interpolation within the training
> augmentation distribution.
>
> However, we agree that no synthetic family can fully cover the space of real-world anomalies. As discussed in Appendix A.10 and A.11, the anomaly generation process is best viewed as an inductive bias informed by domain knowledge, and practitioners are encouraged to adapt the augmentation family to their specific setting. The combination of cross-family generalization (Tables 2–3) and real-world validation (Table 1) provides evidence of practical robustness.

---

> > ### Comment · Reviewer_MJaG · 2026-07-03
> >
> > I thank the authors for the response. My concerns are addressed and I have no further questions.